# The In Vivo Net Energy Content of Resistant Starch and Its Effect on Macronutrient Oxidation in Healthy Adults

**DOI:** 10.3390/nu11102484

**Published:** 2019-10-16

**Authors:** Erin D. Giles, Ian L. Brown, Paul S. MacLean, Zhaoxing Pan, Edward L. Melanson, Kennon J. Heard, Marc-Andre Cornier, Tyson Marden, Janine A. Higgins

**Affiliations:** 1Department of Nutrition & Food Science, Texas A&M University, College Station, TX 77843, USA; 2Australian Cancer Research Foundation, Sydney 2001, Australia; mrbmrsb1@gmail.com; 3Division of Endocrinology, Metabolism, and Diabetes, University of Colorado Anschutz Medical Campus, Aurora, CO 80045, USA; 4Department of Pediatrics, Endocrinology Section, University of Colorado Anschutz Medical Campus, Aurora, CO 80045, USA; 5Department of Biostatistics and Informatics, Colorado School of Public Health, University of Colorado Anschutz Medical Campus, Aurora, CO 80045, USA; 6Division of Geriatric Medicine, University of Colorado Anschutz Medical Campus, Aurora, CO 80045, USA; 7Eastern Colorado Veterans Affairs Geriatric Research, Education, and Clinical Center, Denver, CO 80045, USA; 8Department of Emergency Medicine, University of Colorado Anschutz Medical Campus, Aurora, CO 80045, USA; 9Colorado Clinical and Translational Sciences Institute, University of Colorado Anschutz Medical Campus, Aurora, CO 80045, USA

**Keywords:** resistant starch, net energy, in vivo, fiber, energy expenditure, fat oxidation, carbohydrate oxidation, food labeling

## Abstract

The in vivo net energy content of resistant starch (RS) has not been measured in humans so it has not been possible to account for the contribution of RS to dietary energy intake. We aimed to determine the in vivo net energy content of RS and examine its effect on macronutrient oxidation. This was a randomized, double-blind cross-over study. Eighteen healthy adults spent 24 h in a whole room indirect calorimeter to measure total energy expenditure (TEE), substrate oxidation, and postprandial metabolites in response to three diets: 1) digestible starch (DS), 2) RS (33% dietary fiber; RS), or 3) RS with high fiber (RSF, 56% fiber). The in vivo net energy content of RS and RSF are 2.74 ± 0.41 and 3.16 ± 0.27 kcal/g, respectively. There was no difference in TEE or protein oxidation between DS, RS, and RSF. However, RS and RSF consumption caused a 32% increase in fat oxidation (*p* = 0.04) with a concomitant 18% decrease in carbohydrate oxidation (*p* = 0.03) versus DS. Insulin responses were unaltered after breakfast but lower in RS and RSF after lunch, at equivalent glucose concentrations, indicating improved insulin sensitivity. The average in vivo net energy content of RS is 2.95 kcal/g, regardless of dietary fiber content. RS and RSF consumption increase fat and decrease carbohydrate oxidation with postprandial insulin responses lowered after lunch, suggesting improved insulin sensitivity at subsequent meals.

## 1. Introduction

Resistant starch (RS) is any starch that is not fully digested and absorbed in the upper digestive tract and passes to the large bowel where it is a substrate for microbial fermentation to products including short-chain fatty acids (SCFA; acetate, butyrate, propionate). RS is included in the FDA definition of dietary fiber and, thus, is included in food labels as a non-caloric (0 kcal/g) ingredient in the USA [1]. So, the salvage of energy from the fermentation of RS has not been accounted for in previous dietary research. The most accurate way to assess the amount of energy in a food ingredient that can be used for metabolism and growth is net energy value. Net energy is metabolizable energy minus the heat (i.e., energy) used during digestion of the ingredient, metabolism of nutrients, and excretion of waste [2]. The remaining energy is the only portion of the ingredient available for utilization or storage (see [3] for review). The in vivo net energy value of RS tested in pigs and rats was 83% and 62% of the value for enzymatically degradable starch (DS), or 3.32 and 2.48 kcal/g, respectively [4,5], which is markedly different from the 0 kcal/g used for food labeling in the USA. The in vivo net energy value of RS has not been reported in humans, which may lead to inaccurate available energy calculations in dietary research and misleading food labels for consumers.

RS intake has long been associated with numerous health benefits including decreased postprandial glycemia and insulinemia [6], improved glucose tolerance in hyperinsulinemic men [7], and improved insulin sensitivity in healthy adults [8]. RS intake increases postprandial fat oxidation and decreases postprandial carbohydrate oxidation, although overeating is necessary to cause this effect in a hyperinsulinemic population [9,10]. We have shown that RS intake significantly increases postprandial and 24 h fat oxidation, concomitantly decreasing postprandial carbohydrate oxidation, with no change in daily TEE, in healthy adults [11]. A subsequent study showed that RS alone did not increase postprandial fat oxidation whereas RS plus high protein replicated our previous data [12].

The current study was performed to determine the in vivo net energy value of RS in healthy adults, and to assess the impact of RS consumption on macronutrient oxidation and postprandial metabolite responses to a meal. We hypothesized that the energy content of RS is less than 4 kcal/g, which is the Atwater energy value for DS, but greater than the 0 kcal/g currently being used for food labeling purposes.

## 2. Materials and Methods

### 2.1. Participants

A total of 18 participants (9 male; 9 female) were recruited from the community and the University of Colorado Anschutz Medical Campus. The study was approved by the Colorado Multiple Institutional Review Board (COMIRB) in March 2009, COMIRB protocol number 04-0096, in compliance with the Helsinki Declaration, and the Scientific Advisory and Review Committee of the University of Colorado Hospital (UCH) Clinical Translational Research Center (CTRC). Full written informed consent was obtained from all participants prior to participation in the study. Ethnicity was an optional question on the screening questionnaire; 72% of participants answered this question, all of whom were Caucasian.

To participate, potential participants were required to be between 25 and 45 years of age, have a moderate level of physical activity (no more than 4 one-hour bouts of planned physical activity per week), and a body mass index (BMI) between 20 and 29 kg/m^2^. Participants meeting these criteria underwent a screening visit including an oral glucose tolerance test. Only those with normal glucose tolerance (fasting glucose concentration <6 mM and postprandial glucose concentration not higher than 9 mM) were enrolled in the study. Female participants were taking oral contraceptive pills or progesterone injections and were tested during the early follicular phase of the menstrual cycle. Participants were 28.5 ± 3.8 years of age, 1.54 ± 0.04 m tall, weighed 71.1 ± 12.3 kg, and had a BMI of 23.4 ± 2.1 (mean ± SD).

### 2.2. Study Design

This was a randomized, double blind crossover trial. Participants completed three study visits, approximately 4 weeks apart. Meals during each study visit differed only in starch composition: 1) a totally digestible, 0% RS cornstarch (Amioca, 0% fiber; DS), 2) a cornstarch with 47.6% RS (HiMaize 958, 33% dietary fiber; RS), or 3) a cornstarch with 47.6% RS and high dietary fiber content of 56% (HiMaize 1043; high fiber or RSF). Both RS and RSF were type-2 resistant starch (RS2) and all starches were obtained from National Starch & Chemical Company (Bridgewater, NJ, USA).

Participants received a 3-day energy balance lead-in diet (15% protein, 30% fat, and 55% carbohydrate) prior to each test day, and were required to eat all food provided. Daily energy needs for the lead-in diets were calculated from baseline indirect calorimetry prior to study commencement (resting metabolic rate x daily activity factor of 1.49, using ventilated plexiglass hood for 30 min; Sensormedics 2900 metabolic cart). The lead-in diet was low in total dietary fiber (3 g/1000 kcal/day) to minimize the amount of fermentable material in the bowel on the test day. All diets were designed by CTRC nutritionists using ProNutra software (Viocare Technologies Inc., Princeton, NJ, USA) based on the participant’s food preferences and daily energy needs.

Participants were admitted to the UCH CTRC following a 12 h fast. Gas exchange and total energy expenditure (TEE) were measured in a whole-room calorimeter for a period of 23 h. The 23 h measurement was extrapolated to 24 h to assess daily TEE. An antecubital IV catheter was placed prior to entering the calorimetry room, and all urine during the stay was collected and subsequently analyzed for nitrogen content. Participants consumed a capsule containing 100 mg of Brilliant Blue immediately prior to the first test meal of the day (breakfast) and before bed at the end of the day to act as a fecal marker. Fecal matter was collected from the time of appearance of the Brilliant Blue marker (green stool) and continued until the appearance of the second green stool. Collected feces were kept frozen and returned to the CTRC by the participant. Feces were analyzed for total energy by bomb calorimetry by the University of Alabama Birmingham Nutrition and Obesity Research Center (NORC) Small Animal Phenotyping Core. The gross energy content of the feces was necessary to accurately measure the apparent digestibility of dietary energy as part of the in vivo energy calculations, as previously described [13].

All meals on the calorimetry day consisted of one serving of Jell-O™ (1.5–2 cups, depending on energy needs) that contained the test starch plus regular food items, such as waffles and Canadian bacon or a turkey sandwich, to make up total energy and maintain energy balance. Participants were required to consume all food provided. The regular food items incorporated into the diet were low in fiber to match the fiber content of the 3-day lead-in diet (3 g/1000 kcal), and ensure that the only change from the basal diet was due to the test starch. The diet was 15% protein, 30% fat, and 55% carbohydrate (Table 1); 100% of energy from carbohydrate in the Jell-O™ came from the test starch such that the test starch accounted for 26% of daily energy intake. The Jell-O™ portion of the test meals was distributed throughout the day, with Jell-O™ at breakfast, lunch, and dinner accounting for 10%, 8%, and 8% of total daily caloric needs, respectively. Meals were presented to each participant at baseline (approximately 08, 30), 4.5 h (about 13, 00), and 9 h (about 17, 30). The Jell-O™ portion of the meal was consumed within 1 h of presentation, before remaining regular food items at each meal. Baseline was defined as the time at which the participant began eating the first test meal. The only other intake available over the 23 h stay was water.

The breakfast meal acted as a mixed meal tolerance test. Blood was obtained at baseline and at 30, 60, 90, 120, 180, 240, and 300 min after ingestion of breakfast, where time zero was the time the participant began to ingest the starch. All blood samples were analyzed for glucose, insulin, and triacylglyceride (TAG) concentrations. Glucose, TAG, and urinary nitrogen assays were conducted by the CTRC Core Laboratory using an automated Cobas Mira Plus analyzer (Roche Diagnostics, Basel, Switzerland). Serum insulin measurements were also performed by the CTRC Core Laboratory using a human insulin RIA kit (Linco, St. Louis, LA, USA). Incremental areas under the plasma and insulin curves were calculated as previously described [14].

### 2.3. Whole Room-Indirect Calorimetry

Indirect calorimetry was performed using a previously described indirect calorimetry system (Sable Systems, International, Las Vegas, NV, USA) [15]. O_2_ consumption (VO_2_) and CO_2_ production (VCO_2_) were calculated in 1-min intervals using the flow rate and the differences in CO_2_ and O_2_ concentrations between entering and exiting air, and minute by minute energy expenditure (EE) was calculated using the equations of Jequier et al. [16]. Twenty-four hour VCO_2_ and EE were obtained by summing minute values over the 23 h measurement period and extrapolating to 24 h values. The accuracy and precision of the system was tested monthly using propane combustion tests. The average O_2_ and CO_2_ recoveries during the study were ≥97.0%.

### 2.4. Calculations

Formulae used to calculate non-protein respiratory quotient (RQ; data shown in Appendix A) and subsequent estimations of carbohydrate and fat oxidation were based on the derivations described by Jéquier et al. [16]. The net energy content of the two RS materials was calculated as previously described [13,17].

### 2.5. Statistical Analysis

For this cross-over study, a sample size of 18 participants ensures 80% power at 5% significance if the difference in primary (in vivo net energy value of RS) or secondary (macronutrient oxidation and postprandial metabolite responses) outcomes between two treatment conditions is at least 0.49 times the standard deviation of the difference between study periods within each participant.

To assess the difference of the outcome variables between treatment conditions, linear mixed effects model (LMM) with random subject effect was used to account for the correlations of outcome assessed at different study periods. The LMM model consists three fixed effects: sequence group (i.e., DS-RSF-RS/DS-RS-RSF/RSF-DS-RS/RSF-RS-DS/RS-DS-RSF/RS-RSF-DS), period (i.e., 1st, 2nd or 3rd study visit) and treatment (RS, DS, or RSF). Each fixed effect was treated as a classification variable. Under this model, the least square means adjusted for sequence group and period effects were compared to assess the effect of treatment conditions. All results are expressed mean ± SEM.

Area under curves (AUC) over a 300-min post-prandial interval were calculated using trapezoidal rule for glucose, insulin, and TAG. Treatment effects on these AUC variables were analyzed using the above mentioned LMM model. Moreover, glucose, insulin, and TAG assessed at baseline, 30, 60, 90, 120, 180 and 300 min post-prandially were modeled using the LMM model that includes measuring time and measuring time by treatment in addition to fixed effects of sequence group, period, and treatment. Between-treatment difference at a particular time point was tested using contrasts under this model. Means ± SEM are reported if not otherwise specified. SAS 9.4 software (SAS Institute Inc., Cary, NC, USA) was used for all analyses. *p*-value ≤ 0.05 was deemed statistically significant.

## 3. Results

### 3.1. In Vivo Energy Value

The in vivo energy value for RS and RSF in healthy adults was 2.74 ± 0.41 and 3.16 ± 0.27 kcal/g, respectively. There was no significant difference between the net energy values for RS and RSF (*p* = 0.26) so an average net energy value would be 2.95 kcal/g.

### 3.2. Energy Intake, Expenditure, and Macronutrient Oxidation

Daily energy intake and TEE were not different between diets (Figure 1). Twenty-four hour fat oxidation was higher (100.0 ± 9.19 and 93.3± 6.8 vs. 73.3 ± 7.6 g/day; *p* = 0.02 and 0.05), and carbohydrate oxidation lower (172.0 ± 20.8 and 181.0 ± 21.7 vs. 216.5 ± 20.4 g/day; *p* = 0.05 and 0.06), in response to RS and RSF, respectively, compared to DS (Figure 2). There was no difference in fat or carbohydrate oxidation between RS and RSF. There was no difference in protein oxidation between groups (*p* = 0.66).

### 3.3. Plasma Metabolites

AUC for TAG and glucose concentrations showed no difference between groups (Figure 3A,B). However, insulin AUC was significantly lower for the RS and RSF conditions than DS (6708 ± 493 and 4491 ± 546 vs. 6708 ± 981 µIU/mL·300 min, respectively; RS vs. DS *p* = 0.02, RSF vs. DF *p* = 0.05; Figure 3C). This difference was driven by significantly higher insulin concentration, despite equivalent glucose concentration, in the DS condition (44.5 ± 4.24 µIU/mL) relative to RS and RSF at the 300 min time point only (29.9 ± 4.15 and 29.7 ± 3.95 µIU/mL; *p* = 0.002 for both).

## 4. Discussion

In healthy adults, RS and RSF provided 66% and 74% of the energy provided by ingestion of DS for net energy values of 2.74 ± 0.41 and 3.16 ± 0.27 kcal/g, respectively. This study is the first, to our knowledge, to measure the in vivo energy content of RS in humans and shows that fermentation of RS can salvage energy such that RS cannot be considered a non-caloric food ingredient. Contrary to its current classification by the FDA as a form of dietary fiber yielding 0 kcal/g, RS and RSF could contribute significantly to total daily energy intake. Although the contribution of RS may seem small, simulations from data in China and the USA show that decreasing energy intake by 45–100 kcal/day could prevent age-related weight gain in 90% of the population [18,19]. Using the average RS energy value of 2.95 kcal/day, 100 kcal/day would equate to less than ½ cup Corn Flakes, ¼ cup muesli, 1/3 cup pasta, one piece of white bread, 1/10 cup potato chips, 1/3 of a small corn tortilla, or 1/10 of a banana [20]. Thus, it is feasible that daily energy intake is underestimated if the contribution of RS is not correctly included in food labels.

The finding that resistant starches contribute to net energy also has implications for the design of future clinical research studies. Without accurately knowing the energy value of RS, it has not been possible to account for the contribution of these starches to energy intake, which could confound interpretation of energy balance studies where RS was included as a non-caloric ingredient. In future studies, it would be optimal to use the average energy value of RS (2.95 kcal/g), which is independent of the total dietary fiber of the starch, in all dietary calculations. This is more important for studies taking measurements beyond the 3–4 h postprandial period as fermentation of RS material can only occur after it has reached the bowel, which is 5–11 h in healthy adults [21].

In this study, participants may have been under-fed by up to 65 kcal/1000 kcal, or 130 kcal for a 2000 kcal/day diet, in the RS conditions as all starches were included as 4 kcal/g. This is within acceptable limits for energy balance studies which is considered intake within 150 kcal of TEE per day because the lowest absolute error for whole room indirect calorimetry is 119 ± 16 kcal/day [22]. We compared two different forms of RS, both of which were RS2 subtype and differed only in dietary fiber content. Both RS, at 33% dietary fiber, and RSF, with 56% dietary fiber, increase fat oxidation and decrease carbohydrate oxidation by the same magnitude (carbohydrate oxidation was a trend but did not reach significance for RSF), and have similar in vivo energy values. RS intake was equivalent for both RS and RSF, so it is likely that the RS content of the starch, not the fiber content, is responsible for these physiological effects. Lending credence to this idea, RS has been shown to have adipocyte-specific effects that could contribute to our observed fuel partitioning effects. RS decreases fatty acid synthase (FAS) activity and GLUT4 expression in adipocytes [23] which would act to reduce glucose uptake and availability for storage whereas lower FAS activity would decrease total lipogenesis (lipid synthesis from all carbon sources) in adipocytes. Recently, it has been reported that fermentation of RS to short-chain fatty acids decreases the formation of carbohydrate-derived acetyl-CoA which may contribute to increased postprandial lipid oxidation by relying on fat-derived acetyl-CoA as an alternative fuel source [24] as we have previously postulated

Previous work from our group has demonstrated that replacing 5.4% of carbohydrate in the diet with RS significantly increased dietary fat oxidation, as measured by both ventilated hood indirect calorimetry and radioactive dietary fat tracers [11]. In the current study, we replicated this finding, using whole room indirect calorimetry to show that RS increased fat oxidation by 32% and decreased carbohydrate oxidation by 18%, with no significant change in protein oxidation or total energy expenditure. This lack of effect of RS consumption on energy expenditure would suggest that increasing dietary consumption of RS is unlikely to lead to significant weight loss in healthy adults. However, the ability for RS to alter fuel partitioning (promoting the oxidation of fat rather than storage), suggests that replacing other starches in the diet with RS could have a significant impact on body composition over time, including a decrease in fat mass and preservation of lean mass. Although long-term changes in body composition were not measured in this acute feeding study, chronic RS consumption in rats causes decreased lipogenesis in the white adipose tissue without altering glycogen deposition in liver or muscle [25], which leads to lower weight regain after weight loss on an obesogenic diet, with preservation of lean body mass [26]. The ability to induce such changes in body composition could be particularly appealing for weight maintenance following weight loss and for those at risk of sarcopenia during weight loss, such as the elderly.

The lower insulin areas under the curve for RS and RSF relative to DS were driven by significantly higher insulin concentration, despite equivalent glucose concentration, in DS at the 300 min time point only. The second test meal, lunch, was given at 4.5 h (270 min). Thus, this improvement in insulin sensitivity or action with RS and RSF ingestion that happens at 300 min is indicative of a subsequent meal effect. The subsequent meal effect (or second meal effect) describes the ability of food intake at a single meal to influence postprandial glycemia at the next meal. Although we observed a difference in insulin rather than glucose responses, it is plausible that this effect, like glycemia in the subsequent meal effect, is driven by fermentation of RS. At five hours, this effect is in concordance with data showing that the time frame for fermentation of RS occurs 5–11 h after a meal in healthy adults [21] and concurs with data showing that, at breakfast the day following a dinner containing DS or RS, fermentation of RS caused lower postprandial glycemia [27]. Fermentation of RS produces SCFA which have been previously associated with improved insulin sensitivity and glucose tolerance due to decreased hepatic glucose output and free fatty acid concentrations [28].

There are several limitations of this study that must be considered when interpreting these data. Firstly, this study was conducted in young, healthy adults with a BMI <29 kg/m^2^. Thus, these findings cannot be extrapolated to other populations, including older individuals, or those with impaired metabolic health including those with type-2 diabetes or obesity. Similarly, these findings are specific to RS2 so additional studies will be required to determine the energy content of other RS subtypes. Further, while this study found that the RS content of the starch, not the fiber content, is responsible for the physiological effects, we only tested RS at two levels of fiber content (33% and 56% fiber). Finally, while we observed changes in macronutrient oxidation following consumption of RS in this study, we cannot determine the mechanism/s responsible for these effects.

## 5. Conclusions

In the USA, RS is currently included as 0 kcal/g on food labels. In this study, we demonstrated that the average in vivo net energy content of RS2 in healthy adults is 2.95 kcal/g, regardless of total dietary fiber content of the diet. Thus, changes should be made to food labels to include the energy content of RS. Without such changes, RS may contribute to excess daily energy intake. RS consumption, relative to DS, does not change TEE or protein oxidation but increases fat oxidation and decreases carbohydrate oxidation with insulin sensitivity increased after lunch, in corroboration of the subsequent meal effect. The lack of any effect on TEE precludes the use of RS as a weight loss tool but the ability of RS to alter fuel partitioning suggests that RS could have an impact on body composition over time, causing a decrease in fat mass and preservation of lean mass.

## Figures and Tables

**Figure 1 nutrients-11-02484-f001:**
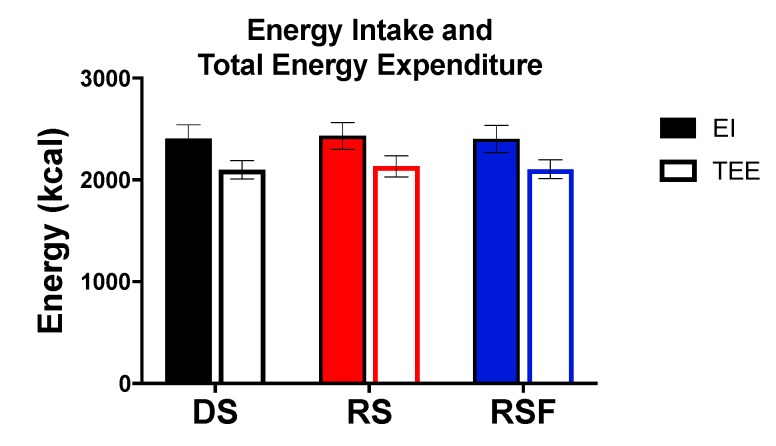
Energy Balance. Energy intake (EI) and total energy expenditure (TEE) during 24 h whole room indirect calorimetry consuming digestible starch (DS), resistant starch (RS), or resistant starch with high fiber (RSF). There are no differences between groups. Data are expressed mean ± SEM.

**Figure 2 nutrients-11-02484-f002:**
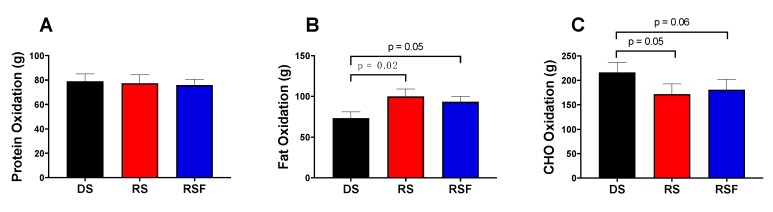
Macronutrient Oxidation. Oxidation of (**a**) Protein, (**b**) Fat, and (**c**) Carbohydrate (CHO) following consumption of digestible starch (DS), resistant starch (RS), or resistant starch with high fiber (RSF), as measured by whole-room indirect calorimetry. Data are expressed mean ± SEM.

**Figure 3 nutrients-11-02484-f003:**
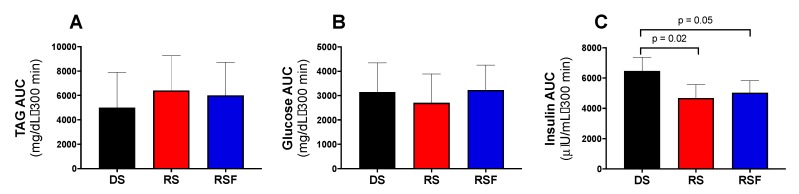
Post-Prandial Plasma Metabolites and Insulin. Change in plasma (**a**) triacylglycerides (TAG), (**b**) glucose, and (**c**) insulin concentrations relative to baseline (fasting) following consumption of a diet containing digestible starch (DS), resistant starch (RS), or resistant starch with high fiber (RSF). There was no effect of diet on incremental area under the curve (AUC) for TAG or glucose, but insulin AUC was significantly lower in RS and RSF relative to DS. Data are expressed mean ± SEM.

**Table 1 nutrients-11-02484-t001:** Diet composition.

	DS (Amioca)	RS (HiMaize 958)	RSF (HiMaize 1043)
Energy Intake (kcal)	2415 ± 141	2392 ± 127	2456 ± 130
Protein (g)	90.3 ± 5.2	89.3 ± 4.7	91.7 ± 4.8
Fat (g)	80.3 ± 4.6	79.2 ± 4.2	81.5 ± 4.3
Carbohydrate (g)	329.5 ± 19.5	327.2 ± 17.3	336.1 ± 17.8
Fiber from test starch (g)	0.0 ± 0.0	52.4 ± 2.8	88.2 ± 4.8
Fiber from Additional Foods (g)	6.4 ± 0.3	6.4 ± 0.3	6.5 ± 0.3

DS, digestible starch, RS, Resistant starch, RSF, RS with high fiber.

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
