# Peer review of "The In Vivo Net Energy Content of Resistant Starch and Its Effect on Macronutrient Oxidation in Healthy Adults"

_nutrients, 2019, doi:10.3390/nu11102484_

Round 1
Reviewer 1 Report
The authors reported a significant calorie intake by resistant starch (RS), in opposition with FDA’s regulation that RS is considered 0 calorie. The authors conducted a pilot clinical study of 18 healthy participants, either sex at young to middle ages, ranging of normal to overweight for a 24h study of total energy expenditure (TEE). They were digestible starch (DS), RS (33%), or RS with high fiber (56%). With the same TEE, same protein oxidation, the groups of RS and RSF showed increased fat oxidation. The average in vivo energy content of RS is 2.95 kcal/g. The authors concluded these effects were due to improved insulin sensitivity that reduced carbohydrate oxidation. And, claimed that the energy contributed by RS should be adequately included in food labels or RS may also contribute to excess daily energy intake.
Mechanism(s) of action(s), such as elevation of insulin sensitivity would need more of animal studies to understand the pathways. Future studies exploring a larger scale of human populations with variety of clinical conditions would be very helpful for nutritional intervention.
The study is a new angle of digging deeply for a full picture of causes of excessive energy intake that may lead to overwight. This study appears to be very thoroughly designed and carefully executed, a randomized, double blind crossover trial containing lead-in diet of 3 days and women follicular phase of the menstrual cycle. Blood, urine and feces (fecal marker) were collected during whole-room indirect calorimeter for a period of 23h after test meal, for analyses of glucose, insulin, free fatty acid, triacylglycerides, and urinary nitrogen. The statistics were appropriate.
The results showing the TEE and protein oxidation did not change, while carbohydrate oxidation was decreased with unaltered glucose levels but reduced insulin levels in the groups of RS and RSF. The changes and their statistical significance were acceptable.
The author claimed that the energy contributed by RS should be adequately included in food labels or RS may also contribute to excess daily energy intake.
Since the RS brought in considerable amount of energy while increase insulin sensitivity, would the authors suggest an appropriate percentage of RS intake?
The abstract was informative. The background briefly summarized the general information of authoritative caloric labeling and a reasonable concern of calorie contribution of RS.
The “methods” section was detailed. The authors provided detailed study procedures including women follicular phase of the menstrual cycle, and adequate information of suppliers for the interested reader to recreate the experiments.
The statistics is adequate and there was information about the software used in this study. .
The results are thoroughly described, and the data were convincingly presented.
The discussion and the conclusions are clear. Mechanism(s) of action(s), such as elevation of insulin sensitivity would need more of animal studies to understand the pathways. Future studies exploring a larger scale of human populations with variety of clinical conditions would be very helpful for nutritional intervention.
Minor edit: in vivo etc. Latin terms needed to be italic.
Reviewer 2 Report
The paper investigated an interesting topic however some criticasl points have to be considered. The main aim of the study is to evaluate the oxidation of macronutrients but no robust determinations were carried out. In order to reach more concrete results I suggest to evaluate lipid oxidation markers such as oxi-LDL, Malondialdehyde and 4-Hydroxy-2-Nonenal, sterol oxidation, oxidized fatty acids. If not how the authors can reach supported conclusions? Also for carbohidrates I suggest to use concrete parameters.
Author Response
We’d like to thank the reviewers for their insightful comments, which have strengthened the paper and clarified certain concepts. Our response to each comment and a summary of the changes made are outlined below:
Reviewer #2 Comment:
The paper investigated an interesting topic however some criticasl points have to be considered. The main aim of the study is to evaluate the oxidation of macronutrients but no robust determinations were carried out. In order to reach more concrete results I suggest to evaluate lipid oxidation markers such as oxi-LDL, Malondialdehyde and 4-Hydroxy-2-Nonenal, sterol oxidation, oxidized fatty acids. If not how the authors can reach supported conclusions? Also for carbohidrates I suggest to use concrete parameters.
Response: Reviewer 2 seems to focus on biochemical markers of carbohydrate and fat oxidation; however, the gold standard of substrate oxidation at the whole-body level is indirect calorimetry or nutrient tracers. Nutrient tracers would be the most appropriate measure for evaluating the oxidation of exogenous substrates (i.e. ingested nutrients). We were specifically interested whole body total fat and
carbohydrate oxidation in this study, rather than exogenous substrates; thus, indirect calorimetry would be considered the most appropriate measure for our purposes. We agree that the biochemical markers suggested by Reviewer 2 would be very useful to evaluate mechanism that mediate the changes observed. We hope to continue this line of research and make such measures in future studies.
Reviewer 3 Report
Major comments.
A power calculation to establish number of participants necessary to test the primary hypothesis is not presented. The limitation of the study should be presented/discussed. For example, the study did not assess energy expenditure of physical activity. The RS was tested at two levels of dietary fiber (33% and 56%) using the RS type that was not characterized (e.g. digestibility, class). Thus, the results should not be generalized to RS of any type with any content of dietary fiber.
Some minor comments.
Line 28: “Eighteen” rather than “18”.
Line 18: “..whole room calorimeter…” – indirect? The nomenclature should be consistent through the manuscript.
Line 82: Inclusion criteria required participants to have “have normal glucose tolerance (response to an oral glucose tolerance test; fasting glucose concentration <6 mM, postprandial glucose concentration not higher than 9 mM)..." It is unclear if the GTT and assessment of the postprandial glucose response were conducted during the screening for participation in this study or in another study. In other words, was this study a separate study or rather a part of a bigger study.
Line 88: Were the differences between males and females tested not only for individual characteristics, but also for the study results?
What was the race/ethnicity of the study participants?
Line 99: It is unclear how long were breaks between the 3 room calorimeter study visits.
Lines 91-93: Unclear what “Amioca” and “HiMaize” are (manufacturer, characteristics, etc.).
Line 101. Information regarding accuracy of the room calorimeter in assessing energy expenditure and substrate oxidation should be provided at this point.
Lines 108-110: It is unclear how the results from the analysis of total stool energy by bomb calorimetry were utilized.
Table 1. It is unclear how the energy content of diet was calculated/analyzed.
Line 116: “…carbohydrate calories…” perhaps “energy from carbohydrates”.
Line 123: Were participants required to eat all food provided?
Line 123: Was the water intake measured?
Line 124:”.. first meal of the day..” perhaps “breakfast”.
Line 125: It is unclear if the blood samples were collected after the ingestion of starch at breakfast only or after breakfast, lunch, and dinner.
Line 127: …triacylglycerides (TAG) concentrations.
Line 133: The authors should consider presenting formulae used in calculations as supplementary online materials.
Line 149: “…mearing’’’” perhaps “measuring”.
Lines 155-157: “was” instead of “is” for consistency.
Line 159: It might be beneficial if the RQ data would be also presented. Also, the authors should consider presenting data used in Figures as tables in supplementary online materials (if allowed).
Line 160: “conditions” perhaps “diets”.
Line 163: “vs” perhaps “compared to”.
Figures. Were vales in Figures 1 and 3 also expressed as mean +/- SEM as they are in Figure 2. This should be also mentioned in the statistical analysis section.
Line 176: “µU/ml.300min” - unclear unit.
Line 253: “..is a caloric food ingredient..” – it should be rephrased.
Lines 255-256: “..RS may contribute to excess daily energy intake.” There is no data presented to support this conclusion.
Line 259: “..fuel partitioning..” no definition is provided.
Lines 260-261”.. impact on body composition over time, causing a decrease in fat mass and preservation of lean mass.” This conclusion is not supported by the data presented.
Author Response
We’d like to thank the reviewers for their insightful comments, which have strengthened the paper and clarified certain concepts. Our response to each comment and a summary of the changes made are outlined below:
Reviewer #3 Comments:
Major comments:
A power calculation to establish number of participants necessary to test the primary hypothesis is not presented.
Response: We have included a sentence at the beginning of the statistical analysis section that includes information on the power calculations for this study.
The limitation of the study should be presented/discussed. For example, the study did not assess energy expenditure of physical activity. The RS was tested at two levels of dietary fiber (33% and 56%) using the RS type that was not characterized (e.g. digestibility, class). Thus, the results should not be generalized to RS of any type with any content of dietary fiber.
Response: We appreciate this suggestion and have added a paragraph on limitations at the end of the Discussion section of the manuscript. Regarding measures of physical activity: Physical activity was not measured in this study because the whole room indirect calorimetry measures total energy expenditure,
which includes activity energy expenditure. Finally, RS type has been included in the methods section (type-2 RS), and we have included in the limitations section that a limitation of this study is the inability to generalize our findings to other starch types.
Minor comments.
Line 28: “Eighteen” rather than “18”. This has been fixed
Line 18: “..whole room calorimeter…” – indirect? The nomenclature should be consistent through the manuscript. Thank-you for noticing this. This has been fixed.
Line 82: Inclusion criteria required participants to have “have normal glucose tolerance (response to an oral glucose tolerance test; fasting glucose concentration <6 mM, postprandial glucose concentration not
higher than 9 mM)..." It is unclear if the GTT and assessment of the postprandial glucose response were conducted during the screening for participation in this study or in another study. In other words, was this study a separate study or rather a part of a bigger study. Glucose tolerance for the subjects was performed during the screening visit and those who did not meet the criteria were not enrolled in this study. The text on lines 85-87 has been updated to clarify this point.
Line 88: Were the differences between males and females tested not only for individual characteristics, but also for the study results? Unfortunately, this study was not powered to detect differences between males and females. We have analyzed our primary outcomes to look for sex-differences, but found no
significant differences in this cohort. We certainly agree that future studies designed to look at sex differences are warranted, and hope to include sex as a variable in future studies.
What was the race/ethnicity of the study participants? Information on race/ethnicity was a voluntary parameter on the screening questionnaire and not all participants provided this information. Of those who did reply to this question, 100% were Caucasian. This information has been included in the
manuscript (Line 82-83).
Line 99: It is unclear how long were breaks between the 3 room calorimeter study visits. This information has been added to the methods section of the manuscript (Section 2.2 – Study Design, lines 102-103).
Visits were approximately 4 weeks apart. This allowed for all females to be measured during the early follicular phase of the menstrual cycle, and a similar timing between study visits for males.
Lines 91-93: Unclear what “Amioca” and “HiMaize” are (manufacturer, characteristics, etc.).
This has been added to the methods (lines 106-107)
Line 101. Information regarding accuracy of the room calorimeter in assessing energy expenditure and substrate oxidation should be provided at this point. We have added this information to the manuscript (lines 160-168)
Lines 108-110: It is unclear how the results from the analysis of total stool energy by bomb calorimetry were utilized. We have updated the manuscript to indicate that “The gross energy content of the feces was necessary to accurately measure the apparent digestibility of dietary energy as part of the in vivo
energy calculations, as previously described [13].” (Lines 126-128)
Table 1. It is unclear how the energy content of diet was calculated/analyzed. We have added text to indicate that this was done using ProNutra software by nutritionists in the CTRC (lines 113-115)
Line 116: “…carbohydrate calories…” perhaps “energy from carbohydrates”. This has been fixed.
Line 123: Were participants required to eat all food provided? Yes, participants were required to consume all food provided on both during the 3-day study lead-in diet and on the study test days. This information has been added to the manuscript. (line 109 and 131-132)
Line 123: Was the water intake measured? No, only urinary output was measured as part of the study.
Line 124:”.. first meal of the day..” perhaps “breakfast”. This has been fixed.
Line 125: It is unclear if the blood samples were collected after the ingestion of starch at breakfast only or after breakfast, lunch, and dinner. This was after breakfast only – this has been clarified in the text (line
144)
Line 127: …triacylglycerides (TAG) concentrations. This has been fixed.
Line 133: The authors should consider presenting formulae used in calculations as supplementary online materials. This could be included if the editor feels that it is warranted; however, we have provided accurate references for all calculations and have performed them exactly as described in the cited papers.
Line 149: “…mearing’’’” perhaps “measuring”. Thank-you - This has been fixed.
Lines 155-157: “was” instead of “is” for consistency. This has been fixed.
Line 159: It might be beneficial if the RQ data would be also presented. Also, the authors should consider presenting data used in Figures as tables in supplementary online materials (if allowed).
We have included a supplementary figure with the RQ data (shown below). We would be happy to put the other data in a table if the editor feels that this is warranted; however this may seem duplicative. (Please see attachment)
Line 160: “conditions” perhaps “diets”. This has been fixed.
Line 163: “vs” perhaps “compared to”. This has been fixed.
Figures. Were vales in Figures 1 and 3 also expressed as mean +/- SEM as they are in Figure 2. This should be also mentioned in the statistical analysis section. Yes, all data are +/- SEM and this has been clarified in the statistical analysis section, as well as the figure legends for Fig 1 &3.
Line 176: “μU/ml.300min” - unclear unit. – This is the common units for noting AUC data for GTT (concentration x time). We have, however, changed the height of the dot (from “.” to a “·” for clarity.
Line 253: “..is a caloric food ingredient..” – it should be rephrased. This section has been re-written (now lines 313-316)
Lines 255-256: “..RS may contribute to excess daily energy intake.” There is no data presented to support this conclusion. We apologize for if this was written to suggest that we had data to support this statement; we have modified the wording to clarify this point (now lines 315-317).
Line 259: “..fuel partitioning..” no definition is provided. We have updated this, now on lines 280-281 of the manuscript.
Lines 260-261”.. impact on body composition over time, causing a decrease in fat mass and preservation of lean mass.” This conclusion is not supported by the data presented. Again, we apologize if this was not clear. We have modified the wording to address this concern (line 283)
Round 2
Reviewer 2 Report
The quality of paper was well improved.
Author Response
Thanks!
Reviewer 3 Report
Thank you very much for responding to my comments.
A few minor to consider:
Line 81: number of participants who responded to the race/ethnicity question (n =??)
Line 83: It is unclear why the age range age limit from 28 to 45. If to decrease heterogeneity, it should be mentioned.
Line 90: The average age was 28.5 +/- 3.8 years. How this corresponds to the limit set in Line 83? Similar issue might be with BMI.
Line 150: 23h and 24 h.
Line 160: primary and secondary outcomes should be listed; perhaps in parentheses?
Line 206: ..Hormones. but results are only for insulin.
Line 282: ..obesity. If so, it should be mentioned in Line 280 that participants had "normal" BMI.
Lines 283-285: the statement is somewhat unclear.
Limitations: it should be mentioned that only 2 levels of fiber were tested.
Author Response
We appreciate the fine eye and thorough review by Reviewer #3. These small changes suggested have all been made to increase the accuracy of this manuscript.
Reviewer #3 Comments:
Line 81: number of participants who responded to the race/ethnicity question (n =??)
We have modified this statement. It now reads: “Ethnicity was an optional question on the screening questionnaire; 72% of participants answered this question, all of whom were Caucasian.”
Line 83: It is unclear why the age range age limit from 28 to 45. If to decrease heterogeneity, it should be mentioned.
Thank you for identifying this typo. It should read 25 to 45, and has beem updated in the text.
Line 90: The average age was 28.5 +/- 3.8 years. How this corresponds to the limit set in Line 83? Similar issue might be with BMI.
Thanks for identifying this inconsistently in the data. We have gone back to the
original data to ensure that the values presented are correct. All demographic data, including average age and BMI are correct; however there were typos in the lower limits of the age and BMI. Ranges should read 25-45 years of age; BMI of 20-29. These have been updated in this version of the manuscript. Our
goal was to recruit healthy young subjects, with normal glucose tolerance.
Line 150: 23h and 24 h.
This has been fixed.
Line 160: primary and secondary outcomes should be listed; perhaps in parentheses?
Good suggestion, thanks. We have added these in parentheses.
Line 206: ..Hormones. but results are only for insulin.
Title has been changed to say Insulin rather than hormones.
Line 282: ..obesity. If so, it should be mentioned in Line 280 that participants had "normal" BMI.
Yes. We have updated this to say “healthy adults with a BMI < 29 kg/m2” (line 285 in this version)
Lines 283-285: the statement is somewhat unclear. Limitations: it should be mentioned that only 2 levels of fiber were tested.
The unclear statement has been removed, and we have updated the limitations to include the fact that only 2 levels of fiber were included.